# Clinical anastomosis leakage and associated factors among patients who had intestinal anastomosis in northwest referral hospitals, Ethiopia

**Chanyalew Worku Kassahun** [1]*, **Senetsuhuf Melekamu**[2], **Mahlet Temesgen Alemu**[2]

**1** Department of Medical Nursing, School of Nursing, University of Gondar, Gondar, Ethiopia, **2** Department of Surgical Nursing, School of Nursing, University of Gondar, Gondar, Ethiopia

* chanyalewworku@gmail.com

**Data Availability Statement:** The data set generated in this study will be available upon reasonable request from the corresponding author. As the data set is from human being, there is

## Abstract

### Background

Clinical anastomosis leakage leads to increased hospital costs, lengths of stay, readmissions, reoperations, and death. Therefore, this study aimed to assess clinical anastomotic leakage and associated factors among patients who had intestinal anastomosis in Northwest referral Hospitals, Ethiopia.

### Method

A retrospective cross-sectional study design was conducted among 411 randomly selected patients. The patient's medical records from February 2017 to February 2020 were used. The date range during which patients' medical records were extracted was from March 1 to June 2020. Patient medical record charts and data extraction sheets were used to collect the data. Data was entered into EPI—DATA version 3.1 and exported into SPSS version 25 for analysis. Binary and multiple logistic regression analysis was used to assess the association between dependent and independent variables. P-value of less than 0.05 and odds ratio with 95% CI were used to declare the presence of association.

### Results

The response rate of the study was 99.8%. Of 411 patients, 38 (9.2%) patients developed clinical anastomotic leakage. Age group 0–10 years (AOR = 6.85 95% CI: 1.742–26.97), emergency presentation (AOR = 3.196 95% CI: 1.132–9.025), and pre-operative comorbid disease (AOR = 7.62 95% CI: 2.804–20.68) were significantly associated with anastomotic leak.

### Conclusions

Clinical anastomotic leakage is higher than the expected rate (4.9%-7.2%). Age, emergency presentation, and comorbidities were associated with clinical anastomotic leak. Hence,

ethical restriction in the data set. The name of the ethics committee that provided ethical clearance is school of nursing ethics committee. In order to maintain the long term data accessibility, you can contact the chair of the school ethical committee with debre2012@gmail.com or dayelegne@gmail.com.

**Funding:** The University of Gondar has funded to conduct the research, but there is no funds for the publication process. The funder had no role in study design, data collection and analysis, decision to publish, or preparation of the manuscript.

**Competing interests:** There is no competing interest.

**Abbreviations:** CAL, Clinical Anastomotic leakage; AOR, Adjusted odds ratio; ASA, American Society of Anesthesiologists; BSC, Bachelors of Science; CAD, Coronary Artery Disease; COPD, Chronic Obstructive Pulmonary Disease; Cr, Creatinine; GP, Generalized peritonitis; GSV, Gangrenous sigmoid volvulus; HCT, Hematocrit; ISK, Ilio sigmoid knotting; LBO, large bowel obstruction; PONV, Perioperative nausea and vomiting; SBO, Small bowel obstruction; SV, Sigmoid volvulus; TASH, Tikur Ambesa Specialized Hospital; TME, Total Mesorecta Excision; UK, United Kingdom; VIF, Variance inflation factor; WHO, World Health Organization.

attention to early identification of risk factors and providing optimal pre-operative, operative, and post-operative care is necessary.

## Introduction

Bowel anastomoses are common surgical procedures in both elective and emergency general surgeries. Although there were some variations in the definition of anastomotic leakage, it was defined as the existence of leakage signs, and confirmed by radiographic examination. The leakage signs included leukocytosis, peritonitis, abdominal pain, tenderness, fever, fecal discharge from the pelvic drain, and pelvic abscess [1]. Anastomotic leakage (AL) has three grades, Grade A anastomotic leakage requiring no active therapeutic intervention, Grade B anastomotic leakage requiring active therapeutic intervention but manageable without re-laparotomy, and Grade C anastomotic leakage which is so-called clinical anastomotic leakage is a severe type of AL requiring re-laparotomy [2].

Clinical anastomotic leakage (CAL) that develops following intestinal surgery is one of the major complications that is terrible to surgeons and nurses. This is because of the increased morbidity and mortality as well as its negative effect on the duration of hospital stay and functional and oncologic outcomes [3].

Intestinal surgery is often associated with a long length of stay (8 days for open surgery and 5 days for laparoscopic surgery), increase rates of surgical site infection by up to 20%, and needs high cost. In the hospital stay for elective colorectal surgery, the incidence of perioperative nausea and vomiting (PONV) may be as high as 80% in patients with certain risk factors. After discharge from colorectal surgery, readmission rates have been noted as high as 35.4% [4].

Although there is improvement in the surgical procedure and experiences, it is still one of the most severe complications. Other than the immediate clinical consequences, such as intra-abdominal abscess, peritonitis, sepsis, and increased in-hospital morbidity, and mortality. It also carries long-term complication, such as impaired pelvic organ function, increased local cancer recurrence, cancer-specific mortality, leads to the permanent stoma and it predisposes to poor prognosis [5].

The most location of intestinal anastomosis was in large intestine (50.9%), and the rates of anastomotic leak vary based on the type of anastomosis with entero-enteric having the lowest leak rate (1–2%) and colorectal/coloanal having the highest (4–26%), Ileocolic (1–4%), colo-colic (2–3%), ileorectal (3–7%), and ileoanal pouch (4–7%) [6]. Surgical—related risk factors such as the location of anastomosis, laparoscopic, and open approaches and handsewn versus stapled anastomoses have been shown to be directly related to the risk of a leak. Patient risk factors such as male gender, diabetes (perioperative hyperglycemia, and elevated hemoglobin A1c), American society of anesthesiologists score $\geq 3$, older age, smoking, serum albumin <4, weight loss, anemia, blood transfusion, chemo-radiation, perianastomotic drain placement, mechanical bowel preparation, tumor size, increased operative time, emergency surgery, pre-operative chemotherapy, intra-operative transfusion were significantly associated with increased risk of AL while pelvic drain resulted in decreased risk of AL [1,7,8].

The clinical anastomotic leak is the most feared form of anastomotic leak after colectomy and requires operative management with the aim of controlling life-threatening sepsis. These kinds of patients are typically septic, with increasing abdominal pain, peritonitis, and altered hemodynamic parameters [9].

Early detections and modification of modifiable risk factors especially in elective settings; cumulative and clinical judgment by the surgeon, optimal preoperative care, better operative techniques, and early identification of leaks using clinical signs with biochemical markers are of principal importance in reducing anastomosis leaks. Assessment of each individual patient's risk for the leak must be made at the time of the operation and decisions regarding anastomosis and use of proximal diversion should be made accordingly [10].

WHO guidelines for safe surgery identify potential standards for enhancements in four areas: safe surgical teams, safe anesthesia, prevention of surgical site infection, and measurement of surgical services [11].

There is limited study on the magnitude and factors associated with anastomotic leak in Ethiopia in the study area in particular. Before forwarding suggestions, knowing the burden and the factors that associated with anastomotic leakage plays a vital role in preventing for its occurrence. Therefore, this study aimed to assess the prevalence and associated factors of clinical anastomosis leakage among patients who had intestinal anastomosis.

## Methods and materials

### Study design, period, and area

A hospital-based cross-sectional study design was used to assess clinical anastomotic leakage and associated factors among patients who had intestinal anastomoses from February 2017 to February 2020 in Northwest Amhara regional State referral Hospitals. The data were extracted from March 1 to June 2020 in three referral hospitals (University of Gondar comprehensive and specialized Hospital, Felege Hiwot comprehensive and specialized hospital, and Debre Markos referral hospital). Three of the referral hospitals performed major operations. On average, 10 major operations were performed per day in each hospital.

### Population

All surgical patients who had intestinal anastomoses surgery from February 2017–February 2020 at Northwest Amhara referral hospitals were considered as source and study populations. Surgical patients who had intestine anastomosis and had incomplete recorded data were excluded.

### Sample size determination, Sampling procedure, and study variables

The required sample size was determined by using a single population proportion formula by taking an assumption $Z\alpha/2 = 1.96$ (type I error), p = proportion of anastomotic leak = 10.8% [10], and d = the margin error between the sample and the population which is 3%.

$$n = \frac{(za/2)^2 \cdot p(1-p)}{d2} \quad n = \frac{(1.96)^2 \cdot 0.108(1-0.108)}{(0.03)2} = 412$$

The total final estimated sample size was 412. From the five referral hospitals in North West Amhara, three hospitals were selected randomly. The total sample size was proportionally allocated for the three Hospitals depending on their load of intestinal anastomosis surgery. This study used a simple random sampling technique. The Medical Record Number (MRN) of study participants was filtered first from the logbook according to their operation time. Then, a serial number was given for the remaining participants, and select each record for study using a computer-generated random number (Fig 1).

Clinical anastomotic leak was the outcome variable and socio-demographic factors (Age, sex, marital status, occupation, history of alcohol and smoking), pre-operative factors

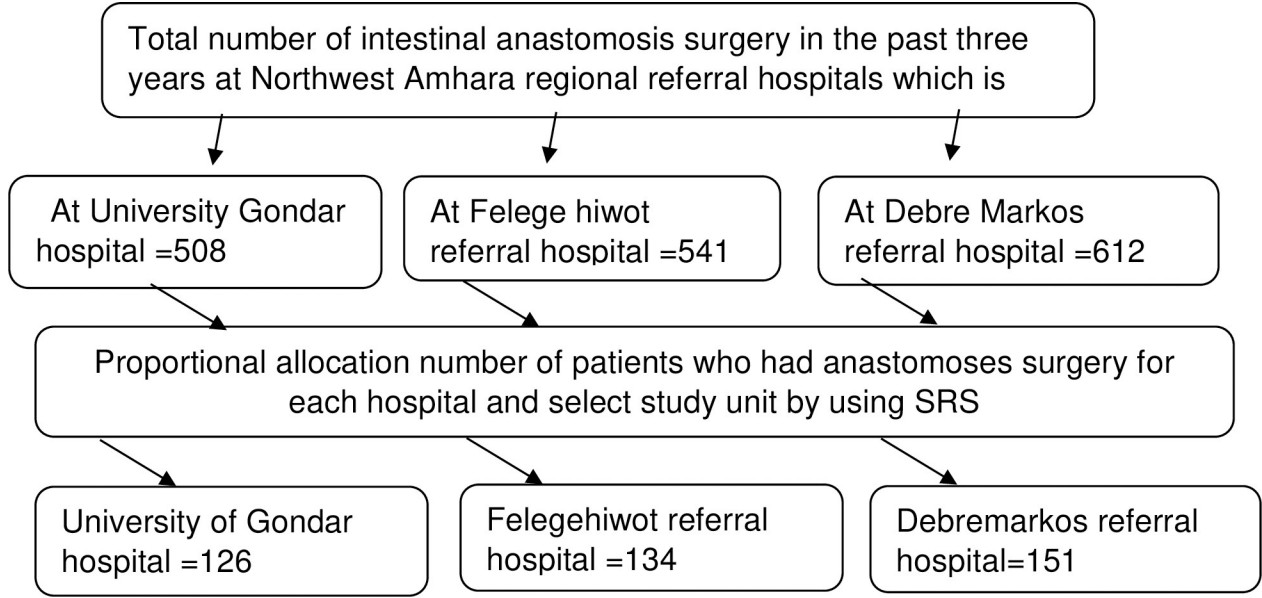

**Fig 1. Schematic presentation of the sampling procedure.**

(comorbid disease, previous surgery, ASA score, diagnosis, location of disease process, presentation, and pre-operative HCT and creatinine), Intra-operative factors (operative time, blood transfusion, type of surgery, type of suture, layer of suture, number of site of anastomosis and use of drain tube) and Post-operative factors (post-operative complication, post-operative hematocrit, Na+, K+, and creatinine level) were independent variables (Fig 2).

## Operational definition

Clinical anastomosis leakage—is defined as a diagnosis of anastomosis leak by the attending physician, and a documented re-operation note or discharge note against medical advice(re-laparotomy) [2]. It can be classified as a low rate of clinical anastomotic leakage if it was less than 4.9%, and a high rate of clinical anastomotic leakage if the rate of leakage was greater than 7.2% [12].

## Data collection instrument, data collection procedures, and quality assurance

Data were collected from patients' charts using a structured checklist. The data extraction sheet is designed based on study objectives and developed by reviewing national and international literature and by observing patient charts. Three supervisors and six data collectors were involved in the data collection process. A 5% preliminary chart review was conducted in the University of Gondar comprehensive specialized referral hospital before the actual data collection. Amendments were considered based on the result of a preliminary chart review. Data collectors and supervisors were trained for one day about the data collection process. Frequent and timely supervision of data collectors was undertaken. The collected data was checked for its completeness during data collection by the principal investigator and supervisor.

## Data processing and analysis

Data were coded and then entered, edited, and cleaned using EPI data version 4.6 and exported to SPSS—25 statistical software for analysis. Descriptive statistics were used to

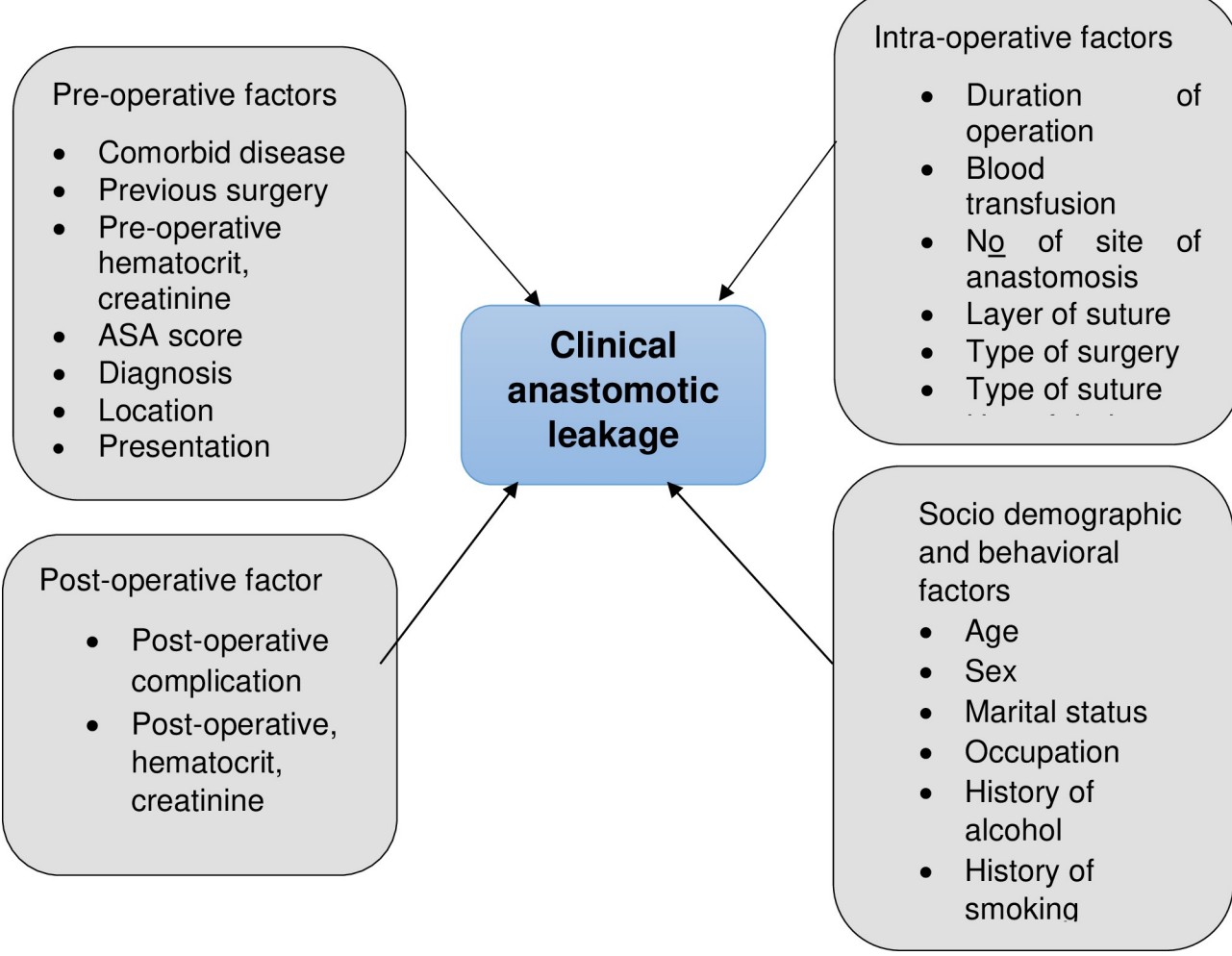

**Fig 2. Relationship between dependent and independent variable.**

describe the socio-demographic characteristics of the respondents, and the magnitude of clinical anastomotic leakage. Text and tables were used to present the findings. The binary logistic regression model was used to assess the association between dependent and independent variables. In multivariable analysis, a P-value of less than 0.05 and an odds ratio with 95% CI were used to declare the presence and the strength of association between the independent and outcome variable. The Hosmer and Lemeshow test was used to diagnose the model's fitness and the model were adequate.

## Ethical approval and consent to participate

Ethical approval was obtained from the ethical review committee of the school of nursing on behalf of the Institutional Review Board (IRB) of the University of Gondar. Permission was obtained from the clinical director of each hospital. As the study uses secondary data, there the need of consent was waived. To ensure confidentiality, personal Identifiers were not used on the data collection form. All data were kept strictly confidential and used only for study purposes. All the data were fully anonymized before the researchers accessed them.

**Table 1.  Socio-demographic and behavioral characteristics among patients who had intestinal anastomosis surgery in Northwest Amhara regional State referral Hospitals, from February 2017 to February 2020. (n = 411).**

| Variables | Response | Frequency | Percent |
|---|---|---|---|
| Age group | 0–10 | 32 | 7.8 |
| | 11–20 | 37 | 9.0 |
| | 21–30 | 67 | 16.4 |
| | 31–40 | 54 | 13.1 |
| | 41–50 | 63 | 15.3 |
| | 51–60 | 66 | 16.1 |
| | > = 61 . | 90 | 21.9 |
| Sex | Male | 280 | 68.1 |
| | Female | 131 | 31.9 |
| Marital status | Single | 90 | 21.9 |
| | Ever married | 321 | 78.1 |
| Occupation | Govt employer | 27 | 6.6 |
| | Farmer | 218 | 53.0 |
| | Merchant | 28 | 6.8 |
| | Student | 47 | 11.4 |
| | Other* | 91 | 22.1 |
| History of alcohol | Yes | 170 | 41.4 |
| | No | 241 | 58.6 |
| History of smoking | Yes | 5 | 1.2 |
| | No | 406 | 98.8 |

* = house wife, driver, NJO, no job, daily laborer.

## Result

### Socio-demographic and behavioral characteristics

A total of 411were participated in the study. Two hundred eighty (68.1%) of them were male and 31.9% (131) of patients were female. About a quarter of the study participants were from the age group > = 61years (21.9%) followed by 21-30(16.4%) with a mean age of 42.5 (S. D ±21.22). The majority of them were ever married 321 (78.1) and farmers 218 (53%). One hundred eight (41.4%) of the participants had a history of alcohol intake and only five of them had a history of smoking (**Table 1**).

### Pre-operative status of the patient related characteristics

Most of the patients had large bowel—related disease (50.9%), and the majority of operation was on an emergency basis (57.4%). In this situation, most of the patients may not have bowel preparation. One hundred seven (26.0%) of them had known chronic co-morbidity and 108 (26.3%) had a history of previous surgery. Pre-operative blood transfusion was given for 34 (8.3%) of the patients, and the majority of the study participants had ASA classes I and II (**Table 2**).

### Intra-operative status of the patient related characteristics

The main form of anastomosis surgeries was primary resection and anastomosis 355 (86.4%) and colostomy/Ileostomy reversal 56 (13.6%). Twenty-six (6.3%) had one additional anastomosis performed. All of the anastomosis was done in open surgery and sutured by hand, single layer in 196 (47.7%) while 215 (52.3%) in double layer. The duration of the procedure was less than three hours in 395 (96.1%) of the patient. Only 36 patients (8.8%) required the intra-

**Table 2. Pre-operative status of the patient related characteristics among patients who had intestinal anastomosis surgery in Northwest Amhara regional State referral Hospitals, from February 2017 to February 2020. (n = 411).**

| Variable | Response | Frequency | Percent |
|---|---|---|---|
| Nature of the disease | Malignant | 26 | 6.3 |
| | Benign | 385 | 93.7 |
| Presentation | Elective | 175 | 42.6 |
| | Emergency | 236 | 57.4 |
| Location | Small intestine | 159 | 38.7 |
| | Large intestine | 209 | 50.9 |
| | Rectum | 12 | 2.9 |
| | Small and large intestines | 31 | 7.5 |
| Indication of anastomosis | LBO 20 SV/ISK | 155 | 37.7 |
| | SBO20POA/volvulus/intussusception | 94 | 22.9 |
| | GP 20 GSV/Injury/perforation | 57 | 13.9 |
| | Colostomy/ileostomy closure | 56 | 13.6 |
| | Others* | 49 | 11.9 |
| Comorbidity | Yes | 107 | 26.0 |
| | No | 304 | 74.0 |
| previous surgery | Yes | 108 | 26.3 |
| | No | 303 | 73.7 |
| Blood transfusion | Yes | 34 | 8.3 |
| | No | 377 | 91.7 |
| HCT | < = 30 | 27 | 6.6 |
| | >30 | 384 | 93.4 |
| creatinine | < = 1.2 | 385 | 93.7 |
| | >1.2 | 26 | 6.3 |
| ASA class | Class I&II | 355 | 86.4 |
| | Class III & IV | 56 | 13.6 |

* = malignancy, intestinal atresia, strangulated inguinal hernia, perforated appendix, ileal cyst.

operative blood transfusion. And the use of drainage tube rate was only 4.9%, and most of them (95.1%) did not have adequate drainage. A total of 56 patients (13.6%) had a history of the stoma (**Table 3**).

## Post-operative status of the patient related factor

Nearly half 193(47%) of the study participants had a recorded post-operative complication, and 118(28.7) of them had history of post-operative blood transfusion. Regarding to the hemo-dynamic characteristics 359 (87.1%) of the patient had a post-operative HCT >30%, and 53 (12.9%) had post-operative HCT<30% (**Table 4**).

## The prevalence of clinical anastomosis leakage

The overall magnitude of clinical anastomosis leakage among patients who had intestinal anas-tomosis was 9.2% with 95% CI (6.6–12.2) (**Fig 3**).

## Factors associated with clinical anastomosis leakage

Multivariable analysis was carried out and three variables were found to be significantly associ-ated with anastomotic leakage. Patients who were in the age group 0–10 were 6.8 times more

 

**Table 3. Intra-operative status of the patient related characteristics among patients who had intestinal anastomosis surgery in Northwest Amhara regional State referral Hospitals, from February 2017 to February 2020. (n = 411).**

| Variable | Response | Frequency | Percent |
|---|---|---|---|
| Type of anastomosis | Primary anastomosis | 355 | 86.4 |
| | Colostomy/ileostomy reversal | 56 | 13.6 |
| No of site of anastomosis | On one site | 385 | 93.7 |
| | On two sites | 26 | 6.3 |
| Intra-operative blood transfusion | Yes | 36 | 8.8 |
| | No | 375 | 91.2 |
| Type of suture | Hand sewn | 411 | 100 |
| Layer of the suture | Single | 196 | 47.7 |
| | Double | 215 | 52.3 |
| Type of surgery | Open | 411 | 100 |
| Use of drainage tube | Yes | 20 | 4.9 |
| | No | 391 | 95.1 |
| Presence of stoma | Yes | 56 | 13.6 |
| | No | 385 | 93.7 |
| Duration of surgery | <3hour | 395 | 96.1 |
| | > = 3hour | 16 | 3.9 |

likely to have clinical anastomosis leakage as compared to age group > = 61 years old (AOR = 6.854,95% CI:1.742–26.968). Patients who underwent surgery as emergency presentation were three times more likely to have clinical anastomosis leakage as compared to patients who underwent surgery as elective presentation (AOR = 3.196, 95%CI: 1.132–9.025). The odds of having clinical anastomosis leakage among patients who had pre-operative chronic co-morbidity 7.6 times higher than their counterparts (AOR = 7.615, 95% CI: 2.804–20.679) (**Table 5**).

## Discussion

In this study, the magnitude of anastomotic leakage was 9.2%. Age group 0–10, Emergency presentation, and those who had the comorbid disease were associated with clinical anastomotic leak.

In the current study, almost 9% of patients had clinical anastomosis leakage. The reported value is consistent with the study finding done in Tikur Ambesa Specialized Hospital (TASH) and Menelik II Memorial Hospital (MIIMH) [10], Sweden [13], and Patna, India [14].

However; the prevalence of clinical anastomosis leakage was found to be lower than in studies done at Al Azhar University Cairo, Egypt [15], in India at BRD medical college [16], and in

**Table 4. Post-operative status of the patient related characteristics among patients who had intestinal anastomosis surgery in Northwest Amhara regional State referral Hospitals, from February 2017 to February 2020.(n = 411).**

| Variable | Response | Frequency | Percentage |
|---|---|---|---|
| Post-operative complication | Yes | 193 | 47 |
| | No | 218 | 53 |
| Post-operative blood transfusion | Yes | 118 | 28.7 |
| | No | 293 | 71.3 |
| Post-operative HCT | < = 30% | 53 | 12.9 |
| | >30% | 359 | 87.1 |
| Post-operative creatinine | < = 1.2 | 372 | 90.5 |

 

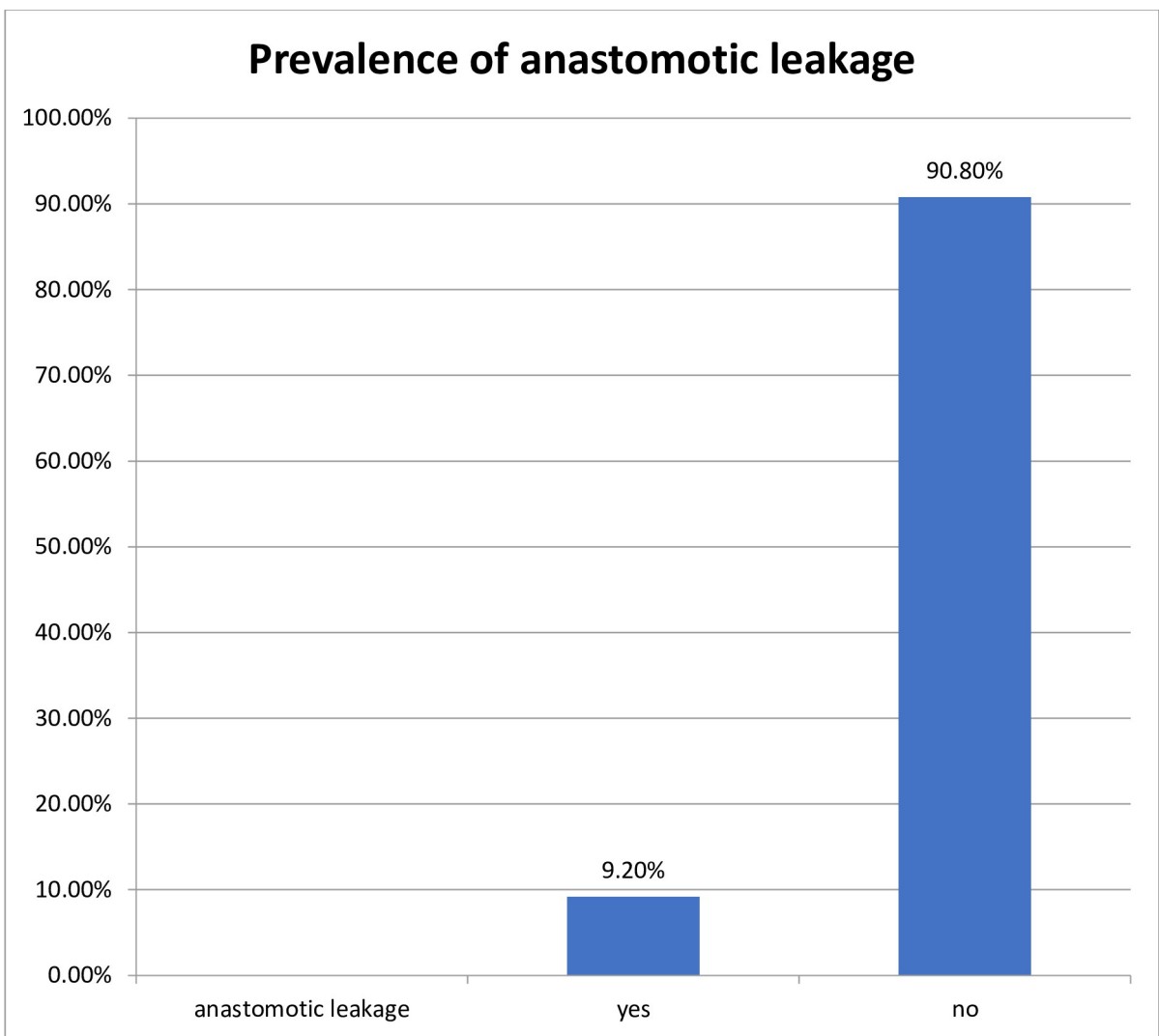

**Fig 3. Magnitude of clinical anastomosis leakage among patients who had intestinal anastomosis in Northwest Amhara regional State referral Hospitals, from February 2017 to February 2020 (n = 411).**

another study in Egypt at Mansoura university hospital [17]. The possible reason for this difference might be due to the fact that the study participants were only primary anastomosis by excluding secondary anastomosis which is colostomy/ileostomy closure. There is also dereference in the indication for anastomosis which was malignancy, and study design difference. Additionally, it is supported by studies in which patients who had primary anastomosis and malignant nature of the disease had a higher rate of leakage [18].

The finding of this study is lower than a study conducted in Global Retrospective Patient Chart Review Study in eight countries [19]. This might be due to the other study considered all grades of anastomotic leakage (grade A, grade B, and grade C) without excluding sub clinical anastomosis leakage.

Nevertheless, it's higher than the finding from the University of Vermont, Burlington, VT [20], Switzerland [21], France [8], Ann Arbor, MI [22], University of Western Australia [23] and New York [24]. This discrepancy might be due to the advanced setting of health service provision and surgical approach in the pre, intra, and post-operation phase they followed

**Table 5. Multivariable logistic regression of factors associated with clinical anastomosis leakage among patients who had intestinal anastomosis in Northwest Amhara regional State referral Hospitals, from February 2017 to February 2020 (n = 411).**

| Variable | Response | clinical anastomosis leakage | | COR 95%CI | AOR 95% CI |
|---|---|---|---|---|---|
| | | yes | no | | |
| Age | 0–10 | 9 | 23 | 3.130(1.137–8.620) | **6.854(1.742–26.968)** ** |
| | 11–20 | 6 | 31 | 1.548(0.519–4.623) | 3.257(0.808–13.134) |
| | 21–30 | 1 | 67 | 0.000(.000-.000) | 0.000(0.000–0.000) |
| | 31–40 | 1 | 53 | 0.151(.019–1.214) | 0.282(0.030–2.626) |
| | 41–50 | 5 | 58 | 0.690(0.224–2.125) | 2.192(0.562–8.545) |
| | 51–60 | 6 | 59 | 0.949(0.341–2.639) | 2.769(0.794–9.655) |
| | > = 61 | 10 | 80 | 1.0 | 1.0 |
| Sex | Male | 28 | 252 | 1.344(0.633–2.857) | 1.118(0.347–3.604) |
| | female | 10 | 121 | 1.0 | 1.0 |
| Marital status | single | 8 | 82 | 1.0 | 1.0 |
| | Ever married | 291 | 30 | 0.386(.192-.776) | 1.521(.148–15.639) |
| Alcohol | Yes | 16 | 154 | .967(0.492–1.901) | 1.347(0.451–4.027) |
| | No | 22 | 219 | 1.0 | 1.0 |
| Presentation | Emergency | 32 | 200 | 6.920(1.542–8.359) | **3.196(1.132–9.025)** ** |
| | Elective | 4 | 173 | 1.0 | 1.0 |
| Chronic co morbidity | Yes | 26 | 81 | 7.811(3.776–16.158) | **7.615(2.804–20.679)** *** |
| | No | 12 | 292 | 1.0 | 1.0 |
| Previous surgery | Yes | 11 | 97 | 1.159(0.554–2.425) | 1.024(0.338–3.100) |
| | No | 27 | 276 | 1.0 | 1.0 |
| Type of anastomosis | Primary | 33 | 321 | 1.070(.308–2.196) | 0.328(0.079–1.370) |
| | Stoma reversal | 5 | 52 | 1.0 | 1.0 |
| Layer of suture | Single | 16 | 180 | 1.0 | 1.0 |
| | Double | 22 | 193 | 1.282(.653–2.519) | 1.154(0.471–2.830 |
| Blood transfusion | Yes | 24 | 94 | 5.088(2.528–10.240) | 2.273(0.922–5.604) |
| | No | 14 | 279 | 1.0 | 1.0 |

* = p value<0.05

** = p value<0.01

*** = p value<0.001, p value of Hosmer and Lemeshow test: 0.994.

which had a great effect on surgical outcomes [25,26]. In the other studies, the surgical approach they used during the intestinal resection and anastomosis was mostly laparoscopic method, and staples to suture the surgical opening and also advanced diagnostic techniques they used in order to early identify leakage before reaching further complications [27,28]. This also might be due to the hospital environment difference. Built environment is contributing factor to the positive surgical outcomes [29]. Similarly, the finding of this study is higher than the finding from St. Paul Hospital Millennium Medical College [12] and Italy [30]. This difference might be in most of their study participants had an elective presentation, which has profound influence to reduce clinical anastomosis leakage [10].

The odds of having clinical anastomosis leakage in age group 0–10 was 6.8 times higher when compared to age group > = 61. This finding was inconsistent with studies done in different countries [4]. This might be due to most of the studies excluding age group 0–10 as study participants. This might be, the age groups 0–10 is known by their alleged fragility, as they represent a special population that is thought to be at higher risk for infection due to their immature immune systems. These patients require special attention with close monitoring during their pre-operative, operative, and post-operative course [4].

According to the finding of this research, patients who had emergency presentation were 3 times more likely to had clinical anastomosis leakage than those who had elective presentation. This is in line with studies done in Addis Abeba [10], India [14], UK [31], and in university of Michigan health system, Ann arbor, MI [22]. This could be the bowel was not fully prepared pre-operatively in case of patients who had emergency presentations, and other preoperative preparations to stabilize patient hemodynamic stability. Bowel preparation to empty the colon from stool to prevent complications of infection and anastomotic leakage and pre-habilitation before intestinal surgery is the preferred technique to lower the incidence of clinical anastomosis leakage which is unlikely done in patients who had an emergency presentations [4].

Patients who had the comorbid disease were more than 7.6 times more likely to have clinical anastomosis leakage than patients who did not have comorbid disease. This finding is consistent with studies done in India [22], the UK [31], New York [24], and Egypt [24]. This might be due to comorbidity is being a major limiting factor in wound healing. It causes the immune cells to function ineffectively, thereby raising the risk of infection, by the mechanism of limiting oxygen- and nutrient-enriched blood flowing to the wound site, delaying the transport of nutrients to the wound site, and by impairing transport of waste products, the dead cells, and deoxygenated blood away from the wound site during all the phases of wound healing [32].

## Strength and limitation of the study

This study incorporates multi-site. Since this study assessed anastomotic leakage and associated factor through retrospective document review, the chart lacks some variables, such as weight, height, level of anastomosis from the anal verge, the experience of the surgeon, the estimated amount of blood loss, albumin level, mechanical bowel preparation, nutrition status, and intra-abdominal infection. So, this study did not show the association of these variables with clinical anastomosis leakage. In addition; the frequency and dose of the utilization of alcohol and smoking were not measured. Rather, we took what was recorded in the history of smoking or alcohol in the patient chart.

## Conclusions

Clinical anastomosis leakage is an important postoperative complication among patients who had intestinal anastomosis. The magnitude of clinical anastomotic leakage was 9.2% which is higher than the accepted rate of anastomotic leak (4.9%-7.2%). Age group 0–10, emergency presentation, and those who had pre-operative co—morbid disease were associated with anastomotic leak

## Acknowledgments

The authors would like to express their deep gratitude to medical directors, head nurses of each hospital for their special contribution by facilitating to have the necessary information.

## Author Contributions

**Conceptualization:** Chanyalew Worku Kassahun, Senetsuhuf Melekamu, Mahlet Temesgen Alemu.

**Data curation:** Chanyalew Worku Kassahun, Senetsuhuf Melekamu, Mahlet Temesgen Alemu.

**Formal analysis:** Chanyalew Worku Kassahun, Senetsuhuf Melekamu, Mahlet Temesgen Alemu.

**Funding acquisition:** Chanyalew Worku Kassahun, Senetsuhuf Melekamu, Mahlet Temesgen Alemu.

**Investigation:** Chanyalew Worku Kassahun, Senetsuhuf Melekamu, Mahlet Temesgen Alemu.

**Methodology:** Chanyalew Worku Kassahun, Senetsuhuf Melekamu, Mahlet Temesgen Alemu.

**Project administration:** Mahlet Temesgen Alemu.

**Resources:** Chanyalew Worku Kassahun, Senetsuhuf Melekamu, Mahlet Temesgen Alemu.

**Software:** Chanyalew Worku Kassahun, Senetsuhuf Melekamu, Mahlet Temesgen Alemu.

**Supervision:** Chanyalew Worku Kassahun, Senetsuhuf Melekamu, Mahlet Temesgen Alemu.

**Validation:** Chanyalew Worku Kassahun, Senetsuhuf Melekamu, Mahlet Temesgen Alemu.

**Visualization:** Chanyalew Worku Kassahun, Senetsuhuf Melekamu, Mahlet Temesgen Alemu.

**Writing – original draft:** Chanyalew Worku Kassahun, Senetsuhuf Melekamu, Mahlet Temesgen Alemu.

**Writing – review & editing:** Chanyalew Worku Kassahun, Senetsuhuf Melekamu, Mahlet Temesgen Alemu.

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
