## [Decision Letter · Decision Letter 0]

14 Aug 2022

PONE-D-22-13405Clinical anastomosis leakage and associated factors among patients who had intestinal anastomosis in northwest Amhara regional state referral hospitals, Ethiopia, 2020PLOS ONE

Dear Dr. Kassahun,

Thank you for submitting your manuscript to PLOS ONE. After careful consideration, we feel that it has merit but does not fully meet PLOS ONE’s publication criteria as it currently stands. Therefore, we invite you to submit a revised version of the manuscript that addresses the points raised during the review process.

Please revise.

We look forward to receiving your revised manuscript.

Kind regards,

Academic Editor

PLOS ONE

Journal Requirements:

2. Thank you for submitting the above manuscript to PLOS ONE. During our internal evaluation of the manuscript, we found significant text overlap between your submission and the following previously published works, some of which you are an author.

- https://www.thieme-connect.com/products/ejournals/abstract/10.1055/s-0038-1677025?lang=de

- https://dfwhcfoundation.org/wp-content/uploads/2018/10/ERAS.pdf

- https://link.springer.com/article/10.1007/s00464-015-4117-x

Please revise the manuscript to rephrase the duplicated text, cite your sources, and provide details as to how the current manuscript advances on previous work. Please note that further consideration is dependent on the submission of a manuscript that addresses these concerns about the overlap in text with published work.

"University of Gondar has funded to conduct the research, but there is no fund for publication process."

7. Please amend either the abstract on the online submission form (via Edit Submission) or the abstract in the manuscript so that they are identical.

Reviewers' comments:

Reviewer's Responses to Questions

**Comments to the Author**

1. Is the manuscript technically sound, and do the data support the conclusions?

Reviewer #1: Partly

Reviewer #2: Yes

2. Has the statistical analysis been performed appropriately and rigorously? 

Reviewer #1: Yes

Reviewer #2: Yes

3. Have the authors made all data underlying the findings in their manuscript fully available?

Reviewer #1: Yes

Reviewer #2: No

4. Is the manuscript presented in an intelligible fashion and written in standard English?

Reviewer #1: Yes

Reviewer #2: No

5. Review Comments to the Author

Reviewer #1: The authors retrospective collected 411 patients to analysis the factors related to anastomosis leakage, and concluded that age, emergency presentation and pre-operative comorbid disease were significantly associated with anastomotic leak.

Here I had some recommendations:

1. The authors did not analysis the nutrition status, intra-abdominal infection, or other modifiable risk factors-such as operative time, use of drainage tube etc. So, they cannot give the readers more important things to prevent anastomosis leakage.

2. Most of their patients were large bowel related disease (50.9%), and majority of operation was on emergency basis (57.4%). in this situation, most of patients may not have bowel preparation. And their use of drainage tube rate was only 4.9%, most of them (95.1%) did not have adequate drainage.

3. In abstract, they use “SPSS version 24”, but in the text, it becames “SPSS 25”.

4. In table 2, the percentage of ASA class I&II and class III&IV seems reverse.

5. In table 5, the authors did not analysis the effect of drainage tube.

6. Repeat sentence in Methods and Materials—Population—(All surgical……in the study)

Reviewer #2: Review comments

Manuscript ID PONE-D-22-13405

Title: Clinical anastomosis leakage and associated factors among patients who had intestinal anastomosis in northwest Amhara regional state referral hospitals, Ethiopia, 2020

Thank you for inviting me to review this interesting research. I read and review the research with great interest. The author tried to describe the Clinical anastomosis leakage and associated factors among patients who had intestinal anastomosis in southwest, Ethiopia. My comments are listed below;

- This manuscript needs to be reviewed for typographical, syntax, spelling, and grammatical errors.

- Please rewrite your title as ‘Clinical anastomosis leakage and associated factors among patients who had intestinal anastomosis in northwest referral hospitals, Ethiopia’. The details are better explained in the methods section.

Introduction

Your introduction section is well written but,

- The authors stated more about scientific/known facts

- didn’t answer well the research question, it needs further elaboration to full fill the scientific merits.

- you do not explain the global and national efforts to tackle the problem, gaps in previous scholars that your study is trying to fill in.

The last paragraph in your introduction said ‘Since there is no study, which shows both the prevalence and factors associated with anastomotic leak in our study setting.’ What does it mean? The sentence is not completed. Moreover, why do you need to stick anastomotic leak in your study setting only? Does every specific setting need research? what is the advancement of new knowledge by this study from http://dx.doi.org/10.4314/ejhs.v29i6.14 ?

Methods

- Please state the total hospitals that perform major operations/surgery in the study setting.

- ‘The data was extracted from March 1 to June,2020……’ please correct English as data were…

Sample size determination, Sampling procedure, and study variables

Show the calculation for the sample size.

The sampling procedure is not clear, the readers left confused.

- First, how three hospitals were selected?

- It is not clear on how many total eligible cases (patient cards/records) were entertained per each hospital?

- The proportion of sample contributed per each hospital?

- I suggest using a diagram for more information regarding

- I have a doubt regarding some explanatory variables you really accessed from secondary data/record patient occupation. Please comment on this. And/or acknowledge your limitations.

- How do you measure the utilization of alcohol and smoking? The dose and frequency of usage matter? In a lifetime, per day weekly, on holydays only…; also amount?

Data processing and analysis

What does it mean you have used SPSS v. 24 in your abstract section and v.25 in the main text?

Discussion

I suggest that discussion section start with a paragraph highlighting the most relevant accomplishments of the research.

This section can be improved as follows.

a. Main findings in the first paragraph

b. Comparison with existing literature

c. Strengths and Weaknesses of the study should be clearly highlighted

d. Conclusion

Generally,

-please, read the complete manuscript and correct typos.

-Please ensure that your manuscript meets PLOS ONE's style requirements, including those for file naming. The PLOS ONE style templates.

Best wishes!

6. PLOS authors have the option to publish the peer review history of their article (what does this mean?). If published, this will include your full peer review and any attached files.

Reviewer #1: **Yes: **Guo-Shiou Liao

Reviewer #2: No

---

## [Author Response · Author response to Decision Letter 0]

27 Aug 2022

Point by point response to Reviewer’s response 

Response to Reviewer #1

S.No Issues raised Reponses

1. The authors did not analysis the nutrition status, intra-abdominal infection, or other modifiable risk factors-such as operative time, use of drainage tube etc. So, they cannot give the readers more important things to prevent anastomosis leakage. Most of their patients were large bowel related disease (50.9%), and majority of operation was on emergency basis (57.4%). in this situation, most of patients may not have bowel preparation. And their use of drainage tube rate was only 4.9%, most of them (95.1%) did not have adequate drainage. • Stated in the limitation as “This study did not assess association of this variables with clinical anastomosis leakage. But, operative time and use of drainage tube have failed to pass the chi-square test assumption, and presented descriptively. 

2. In abstract, they use “SPSS version 24”, but in the text, it becames “SPSS 25”. We used SPSS version -25 for analysis, and modified as SPSS version -25

3. In table 2, the percentage of ASA class I&II and class III&IV seems reverse. Modified as per the recommendation 

4. In table 5, the authors did not analysis the effect of drainage tube. We did not include the effect of drainage tube as it fails to pass the chi-square test assumption.

5. Repeat sentence in Methods and Materials—Population—(All surgical……in the study) Modified 

 

Response to Reviewer#2

S.No Manuscript section Issues raised Reponses

6. General This manuscript needs to be reviewed for typographical, syntax, spelling, and grammatical errors. Modified 

7. Title Rewrite your title as ‘Clinical anastomosis leakage and associated factors among patients who had intestinal anastomosis in northwest referral hospitals, Ethiopia’. Modified as per the recommendation 

8. Introduction The authors stated more about scientific/known facts we tried and modified 

 Didn’t answer well the research question, it needs further elaboration to full fill the scientific merits. we tried and modified 

 You do not explain the global and national efforts to tackle the problem, gaps in previous scholars that your study is trying to fill in. WHO guidelines for safe-surgery identify potential standards for enhancements in four areas: safe surgical teams, safe anesthesia, prevention of surgical site infection, and measurement of surgical services

 The last paragraph in your introduction said ‘Since there is no study, which shows both the prevalence and factors associated with anastomotic leak in our study setting.’ What does it mean? The sentence is not completed. Moreover, why do you need to stick anastomotic leak in your study setting only? Does every specific setting need research? what is the advancement of new knowledge by this study from http://dx.doi.org/10.4314/ejhs.v29i6.14 ?

 • The last paragraph is modified as “There is limited study on magnitude and factors associated with anastomotic leak in Ethiopia in the study area in particular. Before forwarding suggestions, knowing the burden and the factors that associate with anastomotic leakage plays vital role in preventing for its occurrence.”

• As we stated in the statement of problem there is limited evidence on anastomotic leakage in Ethiopia. There is a need to have more evidences to support the actions with evidences. But, every specific setting does not need research. 

9. Methods 

 Please state the total hospitals that perform major operations/surgery in the study setting.

 • Modified as per the recommendation, and highlighted in the tracked document. 

 The data was extracted from March 1 to June, 2020……’ please correct English as data were… • Modified as per the recommendation, and highlighted in the tracked document.

10. Sample size determination, Sampling procedure, and study variables

 Show the calculation for the sample size. 

The sampling procedure is not clear, the readers left confused. 

• First, how three hospitals were selected? 

• It is not clear on how many total eligible cases (patient cards/records) were entertained per each hospital? 

• The proportion of sample contributed per each hospital? 

• I suggest using a diagram for more information regarding 

• I have a doubt regarding some explanatory variables you really accessed from secondary data/record patient occupation. Please comment on this. And/or acknowledge your limitations. 

• How do you measure the utilization of alcohol and smoking? The dose and frequency of usage matter? In a lifetime, per day weekly, on holydays only…; also amount? 

 • Modified as per the recommendation, and highlighted in the tracked document.

• The frequency and dose of the utilization of alcohol and smoking were not measured. Rather, we took what was recorded in the history of smoking or alcohol in the patient chart. We stated it in the limitation of the study. 

11. Data processing and analysis 

 What does it mean you have used SPSS v. 24 in your abstract section and v.25 in the main text?

 • Modified as per the recommendation, and highlighted in the tracked document.

12. Discussion 

 I suggest that discussion section start with a paragraph highlighting the most relevant accomplishments of the research. 

This section can be improved as follows. 

a. Main findings in the first paragraph 

b. Comparison with existing literature 

c. Strengths and Weaknesses of the study should be clearly highlighted

d. Conclusion 

 • Modified as per the recommendation, and highlighted in the tracked document.

---

## [Decision Letter · Decision Letter 1]

12 Sep 2022

PONE-D-22-13405R1Clinical anastomosis leakage and associated factors among patients who had intestinal anastomosis in northwest referral hospitals, EthiopiaPLOS ONE

Dear Dr. Kassahun,

Thank you for submitting your manuscript to PLOS ONE. After careful consideration, we feel that it has merit but does not fully meet PLOS ONE’s publication criteria as it currently stands. Therefore, we invite you to submit a revised version of the manuscript that addresses the points raised during the review process. Please revise. 

We look forward to receiving your revised manuscript.

Kind regards,

Academic Editor

PLOS ONE

Journal Requirements:

Reviewers' comments:

Reviewer's Responses to Questions

**Comments to the Author**

1. If the authors have adequately addressed your comments raised in a previous round of review and you feel that this manuscript is now acceptable for publication, you may indicate that here to bypass the “Comments to the Author” section, enter your conflict of interest statement in the “Confidential to Editor” section, and submit your "Accept" recommendation.

Reviewer #1: All comments have been addressed

Reviewer #2: All comments have been addressed

2. Is the manuscript technically sound, and do the data support the conclusions?

Reviewer #1: Yes

Reviewer #2: Yes

3. Has the statistical analysis been performed appropriately and rigorously? 

Reviewer #1: Yes

Reviewer #2: Yes

4. Have the authors made all data underlying the findings in their manuscript fully available?

Reviewer #1: Yes

Reviewer #2: Yes

5. Is the manuscript presented in an intelligible fashion and written in standard English?

Reviewer #1: Yes

Reviewer #2: Yes

6. Review Comments to the Author

Reviewer #1: The most location of intestinal anastomosis was in large intestine (50.9%), and as we know, rates of anastomotic leak vary based on the type of anastomosis with entero-enteric having the lowest leak rate (1–2%) and colorectal/coloanal having the highest (4–26%). Why did the authors do not place a drainage tube for them, only 4.9%?

Reviewer #2: (No Response)

7. PLOS authors have the option to publish the peer review history of their article (what does this mean?). If published, this will include your full peer review and any attached files.

Reviewer #1: **Yes: **Guo-Shiou Liao

Reviewer #2: No

---

## [Author Response · Author response to Decision Letter 1]

14 Sep 2022

Response to editors #1

 Journal Requirements: We have checked the references , and it is complete and correct. No change is made in the financial disclosure . PACE is used to format the figures.

Response to Reviewer#1

We were unable to do that because we used secondary data. We took what was recorded in the patients’ record. We stated this in the limitation section.

---

## [Decision Letter · Decision Letter 2]

19 Sep 2022

Clinical anastomosis leakage and associated factors among patients who had intestinal anastomosis in northwest referral hospitals, Ethiopia

PONE-D-22-13405R2

Dear Dr. Kassahun,

We’re pleased to inform you that your manuscript has been judged scientifically suitable for publication and will be formally accepted for publication once it meets all outstanding technical requirements.

Kind regards,

Academic Editor

PLOS ONE

Additional Editor Comments (optional):

Reviewers' comments:

Reviewer's Responses to Questions

**Comments to the Author**

1. If the authors have adequately addressed your comments raised in a previous round of review and you feel that this manuscript is now acceptable for publication, you may indicate that here to bypass the “Comments to the Author” section, enter your conflict of interest statement in the “Confidential to Editor” section, and submit your "Accept" recommendation.

Reviewer #1: All comments have been addressed

Reviewer #2: All comments have been addressed

2. Is the manuscript technically sound, and do the data support the conclusions?

Reviewer #1: Yes

Reviewer #2: Yes

3. Has the statistical analysis been performed appropriately and rigorously? 

Reviewer #1: Yes

Reviewer #2: Yes

4. Have the authors made all data underlying the findings in their manuscript fully available?

Reviewer #1: Yes

Reviewer #2: Yes

5. Is the manuscript presented in an intelligible fashion and written in standard English?

Reviewer #1: Yes

Reviewer #2: Yes

6. Review Comments to the Author

Reviewer #1: Conclusions are presented in an appropriate fashion and are supported by the data. No further questions.

Reviewer #2: All comments addresed.

The manuscript is written concisely

Method is adquate

Result and discusion well written

7. PLOS authors have the option to publish the peer review history of their article (what does this mean?). If published, this will include your full peer review and any attached files.

Reviewer #1: No

Reviewer #2: **Yes: **Birhan Tsegaw Taye

---

## [Editor Report · Acceptance letter]

29 Sep 2022

PONE-D-22-13405R2 

Clinical anastomosis leakage and associated factors among patients who had intestinal anastomosis in northwest referral hospitals, Ethiopia 

Dear Dr. Kassahun:

I'm pleased to inform you that your manuscript has been deemed suitable for publication in PLOS ONE. Congratulations! Your manuscript is now with our production department. 

Kind regards, 

on behalf of

Dr. Robert Jeenchen Chen 

Academic Editor

PLOS ONE